



# Effect of freezing on the microstructure of a highly decomposed peat material close to water saturation when used prior to X-ray micro computed tomography

Hassan Al Majou[1,2], Ary Bruand[1] Olivier Rozenbaum[1,3], Emmanuel Le Trong[1]

[1] Université d'Orléans, CNRS, BRGM, Institut des Sciences de la Terre d'Orléans (ISTO), 1A rue de la Férollerie, 45071 Orléans Cedex 2 (France)

[2] University of Damas, Department of Soil Science, Faculty of Agronomy, PO Box 30621, Damas (Syria)

[3] CNRS, Conditions Extrêmes et Matériaux : Haute Température et Irradiation (CEMHTI), UPR 3079, 1 Avenue de la Recherche Scientifique, 45071 Orléans, Cedex 2 (France)

*Correspondence to:* Ary Bruand (Ary.Bruand@univ-orleans.fr)

**Abstract.** The modelling of peatland functioning, in particular the impact of anthropogenic warming and direct human disturbance on $CO_2$, $CH_4$ and $N_2O$, requires detailed knowledge of the peat structure and of both water and gas flow with respect to the groundwater table level. To this end, freezing is nowadays increasingly used to obtain small size peat samples for X-ray micro computed tomography (X-ray μ-CT) as required by the need to increase the resolution of the 3D X-ray CT images of the peat structure recorded. The aim of this study was to analyze the structure of a peat material before and after freezing using X-ray μ-CT and to look for possible alterations in the structure by investigating looking at the air-filled porosity. A highly decomposed peat material close to water saturation was selected for study and collected between 25 and 40 cm depth. Two samples 4×4×7 cm$^3$ in volume were analyzed before and after freezing using an X-ray μ-CT Nanotom 180NF (GE Phoenix X-ray, Wunstorf, Germany) with a 180 kV nanofocus X-ray tube and a digital detector array (2304×1152 pixels Hamamatsu detector). Results showed that the continuity and cross section of the air-filled tubular pores several hundreds to about one thousand micrometers in diameter were altered after freezing. Many much smaller air-filled pores not detected before freezing were also recorded after freezing with 470 and 474 pores higher than one voxel in volume (60×60×60 μm$^3$ in volume each) before freezing, and 4792 and 4371 air-filled pores higher than one voxel in volume after freezing for the two samples studied. Detailed analysis showed that this increase resulted from a difference in the whole range of pore size studied and particularly from a dramatic increase in the number of air-filled pores ranging between 1 voxel (216 10$^3$ μm$^3$) and 50 voxels (10.8 10$^6$ μm$^3$) in volume. Theoretical calculation of the consequences of the increase in the specific volume of water by 8.7% when it turns from liquid to solid because of freezing led to the creation of a pore volume in the organic matrix which remains saturated by water when returning to room temperature and consequently to the desaturation of the largest pores of the organic matrix as well as the finest tubular pores which were water-filled before freezing. These new air-filled pores are those measured after freezing using X-ray μ-CT and their volume is consistent with the one calculated theoretically. They correspond to small air-filled ovoid pores several voxels



in volume to several dozen voxels in volume and to discontinuous air-filled fine tubular pores which were both
detected after freezing. Finally, the increase in the specific volume of water because of freezing appears also be
also responsible for the alteration of the already air-filled tubular pores before freezing as shown by the 3D
binary images and the pore volume distribution.


**1 Introduction**
In many peatland studies, the description of peat physical characteristics is derived from only a few basic metrics
such as porosity, bulk density and humification indexes (Michel et al., 2001; Quinton et al., 2009; Michel, 2015;
Kurnain and Hayati, 2016). However, the short- and long-term modelling of peatland functioning, and in
particular the impact of anthropogenic warming and direct human disturbance on atmospheric $CO_2$, $CH_4$ and
$N_2O$, requires detailed knowledge of the peat structure and of both water and gas flow with respect to the
groundwater table level (e.g. Gharedaghloo et al., 2018; Zhao et al., 2020; Glaser et al., 2021; Muller & Fortunat,
2021; Swinnen et al., 2021; Wiedeveld et al., 2021). To achieve this, X-ray Computed Tomography, which is
widely used in science as a non-invasive technique for the study of internal 2D and 3D structures, appears to be a
promising technique to perform new analyses of the structure of peats and of their physical properties.
Improvements in resolution led to the development of X-ray micro computed tomography (X-ray μ-CT), which
has been applied to peat materials. Kettridge and Binley (2008 and 2011) used X-ray μ-CT to investigate gas
content and peat structure. They studied samples 7.2 cm long and 7.2 cm in diameter with a resolution of 100
μm. Quinton et al. (2009) analyzed the structure and hydraulic properties of peats using X-ray CT. They studied
samples 10 cm long and 6 cm in diameter with a resolution of 45 μm and showed how water contents recorded
in the field were related to the inter-particle pore volume distribution. Using the methodology developed by
Quinton et al. (2009), Rezanezhad et al. (2009 and 2010) studied the influence of pore size geometry on peat
unsaturated hydraulic conductivity by combining X-ray μ-CT and digital image processing. They found that the
large reduction in unsaturated conductivity with depth was essentially controlled by the proportion of air-filled
pores. More recently, Turberg et al. (2014) used X-ray μ-CT to analyze various degrees of disturbance related to
the process of peat extraction, working with large samples 15×15×45 cm³ in volume and a medical X-ray
scanner. 3-D images of regular parallelepipeds 2×2×14 cm³ in volume were recorded with a resolution of 371
μm.
Because of the low consistency of peat materials, and consequently of the possible alteration of the structure
during sub-sampling in the peat blocks collected in the field, several authors used freezing before extraction to
avoid deformation during sub-sampling. This strategy gives small undisturbed samples, making it possible to
increase the resolution of the 3D X-ray μ-CT images recorded. The peat samples collected by Kettridge and
Binley (2008) were frozen soon after collection, and then defrosted prior to their study but the reason for
freezing the samples remains unclear, and appears to have been motivated more by storage conditions than by
the sub-sampling methodology. Quinton et al. (2009) and Rezanezhad et al. (2010) froze peat blocks at −10°C
for 48h before sub-sampling cores 10 cm long and 6 cm in diameter which were extracted using a hollow drill bit



mounted on a drill press. Gharedaghloo et al. (2018) used data from Rezanezhad et al. (2009 and 2010) and
modeled water and solute transport in the pore network of 9.92×9.92×9.92 mm$^3$ samples extracted from X-ray μ-
CT images of the peat materials. They showed that the decrease in the hydraulic conductivity with depth was
related to the reduction in pore radius and increase in tortuosity. Improvements in the X-ray μ-CT technique
have led to an increase in the image resolution, requiring the use of smaller-sized samples; this evolution will
inevitably lead to an increasing recourse to a freezing phase to obtain samples with the appropriate size before
analysis.
The question arises, however, whether the implementation of freezing impacts the evolution of the soil structure
during the passage of water from the liquid to the solid state because of its increase in volume by 8.7% (the
density of ice is 0.92 g mL$^{-1}$ while that of the liquid is 1 g mL$^{-1}$). Working on the effects of freezing on the
physical properties and wettability of highly decomposed peats used as growing media, Michel (2015) showed
that freezing was accompanied by a decrease in bulk density and a marked change in the water retention
properties but the pore structure was not analyzed. Wang et al. (2017) used X-ray μ-CT and showed that non-
uniform volumetric shrinkage, referred to as the freeze-necking phenomenon, was observed in an unsaturated
clay soil in a closed freeze-thaw experiment. Liu et al. (2021) studied the impact of freeze-thaw cycles on the
pore structure characteristics of silty soil using X-ray μ-CT with a 25 μm resolution on the volume of interest,
namely 8.75×8.75×8.75 mm$^3$. Results showed an increase in the macroporosity and pore-throat network
complexity. Ma et al. (2021) studied the effect of freeze-thaw cycles on the pore distribution in soil aggregates 5-
7 mm in diameter using the 3D images with 3.25 μm resolution produced by synchrotron-based X-ray μ-CT.
Results showed how the creation of pores resulting from freeze-thaw cycles can explain changes in the stability
of aggregates.
As the properties of porous materials are controlled by macro- and micro-pore distribution and topology (Vogel,
2002), it is important to pay close attention to the quality of the pore distribution and topology description
resulting from X-ray μ-CT analysis. Since the latter can only be used to study the air-filled pores, those occupied
by water being very difficult to distinguish from the water-saturated organic matrix, the possible alteration of
both the pore network geometry and its saturation degree during sample preparation remains a concern.
However, little attention has been paid to the possible alteration of the pore network during sample preparation
which requires freezing to obtain subsamples with the adequate size prior to X-ray μ-CT analysis. As freezing is
nowadays increasingly used to obtain small size samples of peat materials for X-ray μ-CT analysis, the objective
of this study was to analyze a highly decomposed peat material before and after freezing using X-ray μ-CT to
assess whether freezing modified its structure or not by analyzing the air-filled pores before and after freezing.
**2 Materials and methods**
**2.1 Field sampling**
Highly decomposed *Sphagnum* and *Molinia* peats were sampled in duplicate (samples A and B) in sites which
were intensively studied by D'Angelo et al. (2016), Bernard-Jannin et al. (2018) and Leroy et al. (2018, 2019a
and 2019b). Large undisturbed samples (15×15×25 cm$^3$) were collected between 25 and 40 cm depth when the
groundwater table level was close to the soil surface. The samples were stored at 3–4 °C in sealed plastic bags.



## 2.2 Physico-chemical analysis

Bulk density and particle density were determined by using undisturbed peat samples a few cubic centimeters in volume and the kerozene method developed by Monnier et al. (1973). The total porosity was obtained by dividing the volume of water contained in a saturated sample by the known volume of the sample as described by Boelter (1976) and Nimmo (2013). The water content of the collected samples was determined after oven drying at 105°C for 24h. The degree of peat decomposition was characterized with the pyrophosphate index (Kaila 1956) which was determined following Gobat et al. (1986). The C and N contents were determined by combustion of dried and crushed samples at 1100°C, using a CNS-2000 LECO apparatus.

## 2.3 Sub-sampling in the laboratory

In order to have samples of the appropriate size for X-ray μ-CT, sub-samples of peat materials 4×4×7 cm$^3$ in volume corresponding to the depth of 30–37 cm were prepared by cutting with a scalpel blade to limit disturbance of the peat structure as far as possible. Then, each sample was placed in a transparent plastic tube 5 cm in diameter which was then hermetically sealed to avoid water loss. They were first submitted to X-ray μ-CT and then, on the basis of the methodology developed and used by Rezanezhad et al. (2010), Ramirez et al. (2016) and Moore et al. (2017), they were frozen at -10°C for 48 h, defrosted for 48 h at 20°C and submitted again to X-ray CT. Each sealed plastic tube with its peat material was weighed at the different steps of the process to check the absence of water loss. Measurements showed that the weight variation between two successive steps and between the first and last step was <0.1 g for the two samples studied.

## 2.4 X-ray Computed Tomography imaging (2D and 3D images)

X-ray μ-CT was performed for the sub-samples 4×4×7 cm$^3$ in volume cut between 30 and 37 cm depth using a micro X-ray μ-CT device Nanotom 180NF (GE Phoenix|x-ray, Wunstorf, Germany). This equipment has a 180-kV nanofocus X-ray tube and a digital detector array (2304×1152 pixels, Hamamatsu detector). Samples were placed in the chamber and rotated by 360 degrees during acquisition. The resulting projections were converted into a 3D image stack using a microcluster of four personal computers (PCs) with the Phoenix 3D reconstruction software (filtered back projection Feldkamp algorithm (Feldkamp et al., 1984)). The reconstruction software contains several different modules for artifact reduction (beam hardening, ring artifacts) to optimize the results. Finally, the 16-bit 3D image was converted into an 8-bit image (256 grey levels) before preprocessing. The samples were mounted and waxed on a glass rod. An operating voltage of 110 kV and a filament current of 59 μA were applied. The distance between the X-ray source and the sample and between the X-ray source and the detector was 300 and 350 mm, respectively, giving a voxel size of 60 μm. The 2000 projection images (angular increment of 0.18°) were acquired during stone rotation (with an acquisition time of 4 hours). As the cone beam geometry created artifacts, the first and the last cross-sectional images were removed (Le Trong et al., 2008; Rozenbaum and Rolland du Roscoat, 2014).

The resulting 3D images were cropped for sample A to a size of 430×600×800 voxels corresponding to 2.6×3.6×4.8 cm$^3$ before and after freezing, and for sample B to a size of 430×530×850 voxels before and after



freezing corresponding to 2.6×3.2×5.1 cm$^3$, each image in a local 3D coordinate system with a voxel size of
60×60×60 μm$^3$ for samples A and B before and after freezing.
**2.5 X-ray image analysis (segmentation and attenuation)**
A region of interest that excluded the irregular sample boundaries and outside region was defined for each
sample in the following manner. For each stack of 2D images before freezing, upper and lower slices, well
inside the sample, that contained clearly identifiable features were identified. In the images after freezing, the
slices containing these features were sought. The other slices were discarded. This sets the height of the 3D
images. Each image was then horizontally cropped so as to keep only the interior of the samples. The cropped
region was defined again with respect to clearly identifiable features in the images before and after freezing.
Smoothing the 3D images with a moving average filter over a window of 5×5×5 voxels increased their signal-to-
noise ratios from the range [7.6-9.2] to [12.5-15.0]. They were then segmented by thresholding. The threshold
value used was the absolute minimum between the two peaks of the bimodal distribution of the grey levels of the
voxels of each image (Fig. 1) (Rozenbaum et al., 2012). The grey level corresponding to that threshold value for
sample A before and after freezing was 93 and 78, respectively. For sample B before and after freezing, it was 80
and 68, respectively. This simple procedure has no adjustable parameter and therefore introduces no bias when
comparing the images. In each binary image, each pore (i.e. group of contiguous foreground voxels surrounded
by background voxels) was identified by scanning the image, and its volume (in terms of number of voxels)
recorded.
**3 Results and discussion**
**3.1 Characteristics of the peat samples studied**
The measured physical characteristics of the peat samples studied are given in Table 1. The volumetric water
contents at sampling were similar for the two samples A and B (0.893 and 0.883 cm$^3$ cm$^{-3}$, respectively). These
values are much higher than those recorded by Rezanezhad et al. (2010) for sphagnum peat materials (between
0.38 and 0.43 cm$^3$ cm$^{-3}$) collected between the surface and 67 cm depth. Their peat materials were collected far
from water saturation because the groundwater table level was far from the surface, whereas our samples were
collected with a groundwater table level close to the soil surface. The porosity values of samples A and B (0.918
and 0.904, respectively) are close to their water content, thus indicating that they are close to water saturation.
The measured bulk densities recorded for the two samples A and B (0.135 to 0.178 g cm$^{-3}$, respectively) are
consistent with those of highly decomposed peat materials (Benscoter et al., 2011; Kurnain and Hayati, 2016).
The measured pyrophosphate index recorded for samples A and B (96.1 and 78.9, respectively) are also
consistent with highly decomposed peat materials which can be classified as asapric peat (pyrophosphate index
>30) according to Levesque et al. (1980). The C/N ratio recorded for samples A and B (12.1 and 16.6,
respectively) confirms that the two peat samples present a high degree of decomposition (Comont et al. 2006).
Finally, the dry bulk density values recorded for the two samples A and B (0.135 and 0.178 g cm$^{-3}$, respectively)
are much closer to the values recorded for a well-decomposed peat material resulting from *Sphagnum* moss with



a fiber content of only 15% (0.25 g cm$^{-3}$) than to the values recorded for a similar undecomposed peat material
with a fiber content of 98% (0.009 g cm$^{-3}$) (Boelter, 1968).

**3.2 Comparison of the 2D and 3D X-ray μ-CT images in grey levels before and after freezing**

The same heights were chosen for the 3D X-ray μ-CT images in grey levels before and after freezing for samples
A (800 voxels) and B (850 voxels). The final image sizes chosen were then 430×600×800 voxels
(~2.6×3.6×4.8 cm$^3$) before and after freezing for sample A and 430×530×850 voxels (~2.6×3.2×5.1 cm$^3$) before
and after freezing for sample B.
The grey level on the images was determined by the absorption of the incident X-ray radiation by the different
phases of the peat material. The absorption of each phase depends on its density and mean atomic number
resulting from its chemical composition (Youn et al., 2015). It is described by the Beer-Lambert Law:
$I = I_o \, exp(-\mu x)$ (1)
where $I$ is the transmitted X light, $I_o$ the incident X light, $\mu$ the absorption coefficient, and $x$ the path length.
Consequently, the intensity of the transmitted X light which results in a grey level of the pixel in the 2D images
and of the voxel in the 3D images depends on the proportion of air, water and organic compounds in the pixel or
voxel considered. Because of the weak difference between the mean atomic number assumed for the porous
organic matrix of a highly decomposed and water-saturated peat material (Table 1) and the mean atomic number
of the water phase, we can assume that the absorption coefficient of these two phases is very close. Therefore,
only the air phase can be distinguished from the other phases. Thus, only the air-filled pores are identifiable on
the 2D and 3D images; the pores occupied by water are undistinguishable from the water-saturated porous
organic matrix.
Pairs of 2D X-ray μ-CT images recorded before and after freezing were selected within the pairs of stacks of 2D
images before and after freezing by identifying the closest images in terms of morphology of air-filled pores a
few hundred micrometers in size. These pores are shown in black in Fig. 2. The dark grey background
corresponds to the highly decomposed organic material and related micro-porosity which was filled by water.
For each pair of 2D X-ray μ-CT images, comparison showed the presence of pores recognizable on the images
before freezing which were still present after freezing but exhibiting a different morphology, of pores
recognizable on the images before freezing which were not present after freezing, and the presence of pores
recognizable after freezing and which were not present before freezing. However, the use of pairs of 2D X-ray μ-
CT images does not enable an accurate estimation of the possible evolution of the porosity of peat materials
during the freezing process since it was not possible to say whether the pairs of 2D images corresponded exactly
to the same slice in the sample before and after freezing. Only a 3D analysis is able to establish whether the
porosity of the peat materials is different before and after freezing.

**3.3 Comparison of 3D CT binary images before and after freezing**

The 3D X-ray μ-CT binary images of the two samples A and B were first morphologically compared globally by
comparing the porosity characterized in X-ray μ-CT before and after freezing (Figs. 3a and d, 4a and d). Results
showed that the air-filled pores measured corresponded to a very small proportion of the total porosity of the



peat material studied, less than 0.02, whereas the total porosity of samples A and B was 0.918 and 0.904 before
freezing, respectively (Tables 1 and 2). Most of the porosity corresponded to both water-filled pores associated
to the highly decomposed organic compounds and potentially to larger water-filled pores occupied by water and
consequently indistinguishable from the porous organic matrix.
The number of air-filled pores composing the very small proportion of the total porosity described with the X-
ray μ-CT used was however very different before and after freezing for the two samples studied. There were 470
and 474 air-filled pores before freezing, and 4792 and 4371 air-filled pores after freezing for samples A and B,
respectively (Table 2). Whatever the origin of the new air-filled pores, results showed a strong decrease in the
average size of the air-filled pores after freezing, from 3952 to 732 voxels and from 2043 to 488 voxels for
samples A and B, respectively (Table 2).
Analysis of the pore size distribution showed that the increase in the number of air-filled pores was mainly
related to an increase in the number of pores <500 voxels in volume (i.e. <0.108 mm$^3$) (Figs. 5a, b, c and d). Air-
filled pores >500 voxels were also highly affected (Figs. 5a', b', c' and d'). After separation of the air-filled
pores larger and smaller than 500 voxels in volume, the 3D X-ray μ-CT images showed that the morphology of
the air-filled pores >500 voxels was however affected, with alterations in both their continuity and transversal
section size (Figs. 3c and f, 4c and f). Analysis of the distribution of the pores <500 voxels in volume showed a
strong increase in the number of pores in all sizes, with the highest increase recorded for pores ranging from 1 to
50 voxels in volume (Fig. 6).
Sub-images of the 3D X-ray μ-CT images recorded were selected to analyze the difference in pore morphology
before and after freezing more easily than with the whole images, in which the high number of pores limited the
morphological analysis (Figs. 3 and 4). One 3D X-ray μ-CT sub-image 200×350×350 voxels in volume
(~1.2×2.1×2.1 cm$^3$) and another one 300×300×300 voxels in volume (~1.8×1.8×1.8 cm$^3$) were selected for
samples A and B, respectively (Figs. 7 and 8). The selected sub-images showed that the pores <500 voxels
corresponded to air-filled ovoid pores of several voxels to several dozen voxels, and to discontinuous air-filled
fine tubular pores (Figs. 7b and e, 8b and e). Comparison of the 3D X-ray μ-CT sub-images selected showed that
freezing led to a dramatic increase in the number of air-filled ovoid pores and to the appearance or disappearance
of discontinuous air-filled fine tubular pores (Figs. 7 and 8).
**3.3 Origin of the difference recorded before and after freezing**
As freezing leads to an 8.7% increase in the specific volume of the water, the possible consequences of this
increase on the changes recorded for the peat material studied were analyzed. The total porosity before freezing
($\phi_{T, BF}$) can be written as follows:
$$\phi_{T,BF} = V_{V, BF} / (V_S + V_{V,BF}) \qquad\qquad (2)$$
where $V_{V, BF}$ is the total specific volume of pores of the peat material before freezing in cm$^3$ g$^{-1}$, $V_S$ is the specific
volume of the organic solid phase dried at 105°C in cm$^3$ g$^{-1}$ and equal to 0.591 cm$^3$ g$^{-1}$ and 0.562 cm$^3$ g$^{-1}$ for
samples A and B, respectively (reciprocal of the particle density measured for peat materials A and B) (Table 1).
Thus, using equation (2):
$$V_{V, BF} = \phi_{T, BF} \times V_S / (1 - \phi_{T, BF}) \qquad\qquad (3)$$



which gives $V_{V,\,BF}$ = 6.616 cm$^3$ g$^{-1}$ and 5.292 cm$^3$ g$^{-1}$ for samples A and B, respectively. The specific volume of
pores before freezing, $V_{V,\,BF}$, can be decomposed as follows:
$V_{V,\,BF} = V_{V,\,Mwf,\,BF} + V_{V,\,TPwf,\,BF} + V_{V,\,TPaf,\,BF}$ (4)
where $V_{V,\,Mwf,\,BF}$ is the specific volume of pores of the organic matrix saturated with water before freezing in
cm$^3$ g$^{-1}$, $V_{V,\,TPwf,\,BF}$ is the specific volume of tubular pores occupied with water before freezing in cm$^3$ g$^{-1}$ and
$V_{V,\,TPaf,\,BF}$ is the specific volume of tubular air-filled pores before freezing in cm$^3$ g$^{-1}$. The porosity related to the
tubular air-filled pores before freezing, $\phi_{TPaf,\,BF}$, in the whole peat material is:
$\phi_{TPaf,\,BF} = V_{V,\,TPaf,\,BF} / (V_S + V_{V,\,BF})$ (5)
Thus:
$V_{V,\,TPaf,\,BF} = \phi_{TPaf,\,BF} \times (V_S + V_{V,\,BF})$ (6)
which gives $V_{V,\,TPaf,\,BF}$ = 0.065 cm$^3$ g$^{-1}$ and 0.029 cm$^3$ g$^{-1}$ for samples A and B, respectively, with the values of
$\phi_{TPaf,\,BF}$ corresponding to the value of $\phi$ measured before freezing using X-ray μ-CT (Table 2).
The increase in the specific volume of water by 8.7% because of freezing increases the porosity related to both
the pores of the organic matrix and tubular pores which were occupied with water before freezing to a value of
porosity after freezing which can be calculated as follows:
$\phi_{Mwf,\,AF} + \phi_{TPwf,\,AF} = [(V_{V,\,Mwf,\,BF} + V_{V,\,TPwf,\,BF}) \times 1.087] / [V_S + ((V_{V,\,Mwf,\,BF} + V_{V,\,TPwf,\,BF}) \times 1.087) + V_{V,\,TPaf,\,BF}]$ (7)
where $\phi_{Mwf,\,AF}$ is the porosity related to the water-filled of the organic matrix after freezing, $\phi_{TPwf,\,AF}$ is the
porosity related to the water-filled tubular pores after freezing. Using equation (4):
$\phi_{Mwf,\,AF} + \phi_{TPwf,\,AF} = [(V_{V,\,BF} - V_{V,\,TPaf,\,BF}) \times 1.087] / [V_S + ((V_{V,\,BF} - V_{V,\,TPaf,\,BF}) \times 1.087) + V_{V,\,TPaf,\,BF}]$ (8)
which gives $\phi_{Mwf,\,AF} + \phi_{TPwf,\,AF}$ = 0.916 and 0.906 for peat materials A and B, respectively by using values of
$V_{V,\,BF}$ and $V_{V,\,TPaf,\,BF}$ given by equations (3) and (6), respectively. These values can be compared to those of
$\phi_{Mwf,\,BF} + \phi_{TPwf,\,BF}$ before freezing that were calculated as follows:
$\phi_{Mwf,\,BF} + \phi_{TPwf,\,BF} = \phi_T - \phi_{TPaf,\,BF}$ (9)
with $\phi_{TPaf,\,BF}$ equal to $\phi$ before freezing (Table 2), which gives $\phi_{Mwf,\,BF} + \phi_{TPwf,\,BF}$ = 0.909 and 0.899 for samples A
and B, respectively. Thus, according to these results and the related assumptions, the increase in the water-filled
porosity of samples A and B after freezing was 0.007. These values can be compared with the increase in the air-
filled porosity and measured with the 3D X-ray μ-CT images recorded in this study. This increase after freezing
was 0.008 and 0.006 for samples A and B, respectively and thus similar to the calculated values.
Based on these different results, a scenario can be proposed to explain what happened during the freeze-thaw
process. During freezing, the water-filled pore volume corresponding to the sum of the pores of the organic
matrix and of fine tubular pores increases by 8.7% because of the increase in the specific volume of water when
it turns from liquid to solid. Once returned to room temperature, the peat material keeps the memory of this
evolution during the freezing phase. As the specific pore volume of the highly decomposed organic matrix
increases in volume following the formation of ice, it does not then decrease after thawing, with the result that
the water, which is located preferentially in the smallest pores, small tubular pores and largest pores of the





organic matrix, all saturated with water before freezing, is no longer located in these pores when the water turns
from solid to liquid after thawing. The porosity newly occupied by air was measured using 3D X-ray μ-CT
(Table 2) and corresponds to the increase in porosity calculated following the transformation of liquid water into
ice. The increase in the specific volume of water because of freezing may also be responsible for the alteration of
the already air-filled tubular pores before freezing as shown by the 3D binary images (Figs. 3 and 4) and the pore
volume distribution (Figs. 5 and 6) because of deformations of the organic matrix structure during freezing.
**4. Conclusions**
Our results show that the freezing technique that can be used prior to peat material sub-sampling as required by
3D X-ray CT altered the structure of the highly decomposed and close to water saturation peat material studied.
The tubular pores from several hundreds to about one thousand micrometers in diameter were indeed altered,
with both their continuity and cross section being different before and after freezing. These pores were several
hundred to several thousand voxels in volume in the 40 cm$^3$ in volume highly decomposed peat material studied,
one voxel corresponding to 216 μm$^3$ in volume. Results show also that very small air-filled ovoid pores several
voxels to several dozen voxels in volume and discontinuous air-filled fine tubular pores within the peat material
studied were only detected after freezing. Theoretical calculation of the consequences of the increase in the
specific volume of water by 8.7% when it turns from liquid to solid state because of freezing led to the creation
of a pore volume in the organic matrix which remained saturated by water when returning to room temperature
and consequently to the desaturation of the largest pores of the organic matrix as well as the finest tubular pores
which were water-filled before freezing. These new air-filled pores are those measured after freezing using X-
ray μ-CT and their volume is consistent with the one calculated theoretically. We conclude that the increase in
the specific volume of water because of freezing is also responsible for the alteration of the already air-filled
tubular pores before freezing, as shown by the 3D binary images and the pore volume distribution, and that this
alteration is a consequence of the deformation of the organic matrix due to the increase in the specific volume of
water when it turns from liquid to solid because of freezing. Finally, our results show clearly that both the pore
morphology and pore size distribution, and more globally the structure of the highly decomposed peat material
studied, were altered by freezing. Thus, the use of freezing prior to any study of the structure of peat materials
similar to the one studied here and close to water saturation should be avoided.
*Financial support.* This research was supported by the Labex Voltaire (ANR-10-LABEX-100-01) and the
French program PAUSE.
*Acknowledgements.* The authors acknowledge Dr. Sébastien Gogo for his assistance during field sampling,
Marielle Hatton for her contribution to chemical analysis and Philippe Penhoud for his contribution to X-ray μ-
CT image acquisition.

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






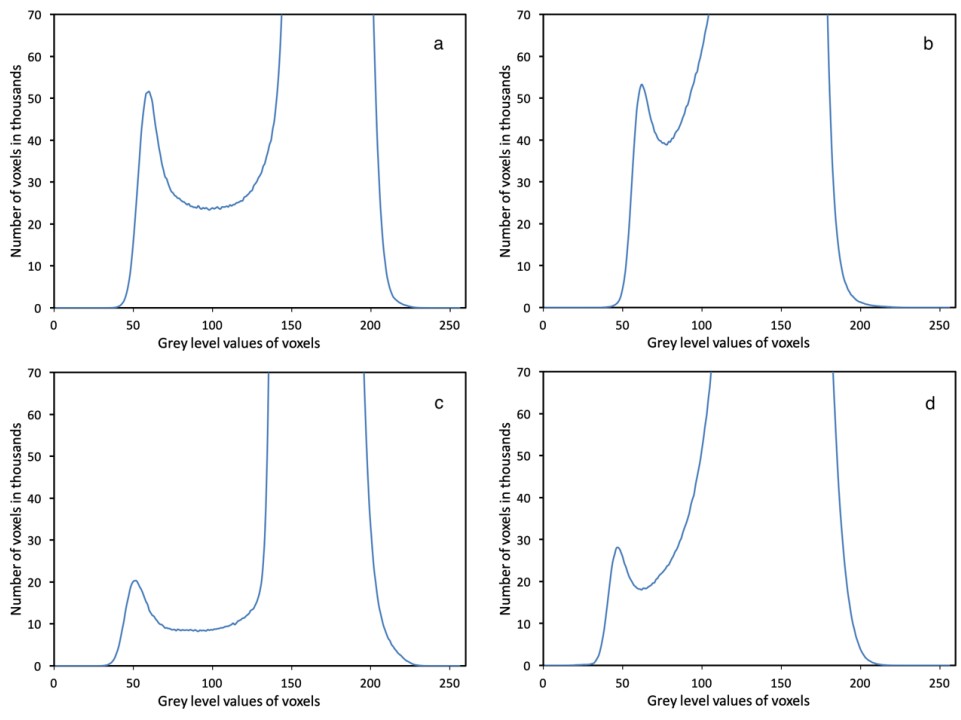


**Figure 1** Distribution of the grey level values in the 3D X-ray μ-CT images recorded for sample A before (**a**)
and after (**b**) freezing and for sample B before (**c**) and after (**d**) freezing.



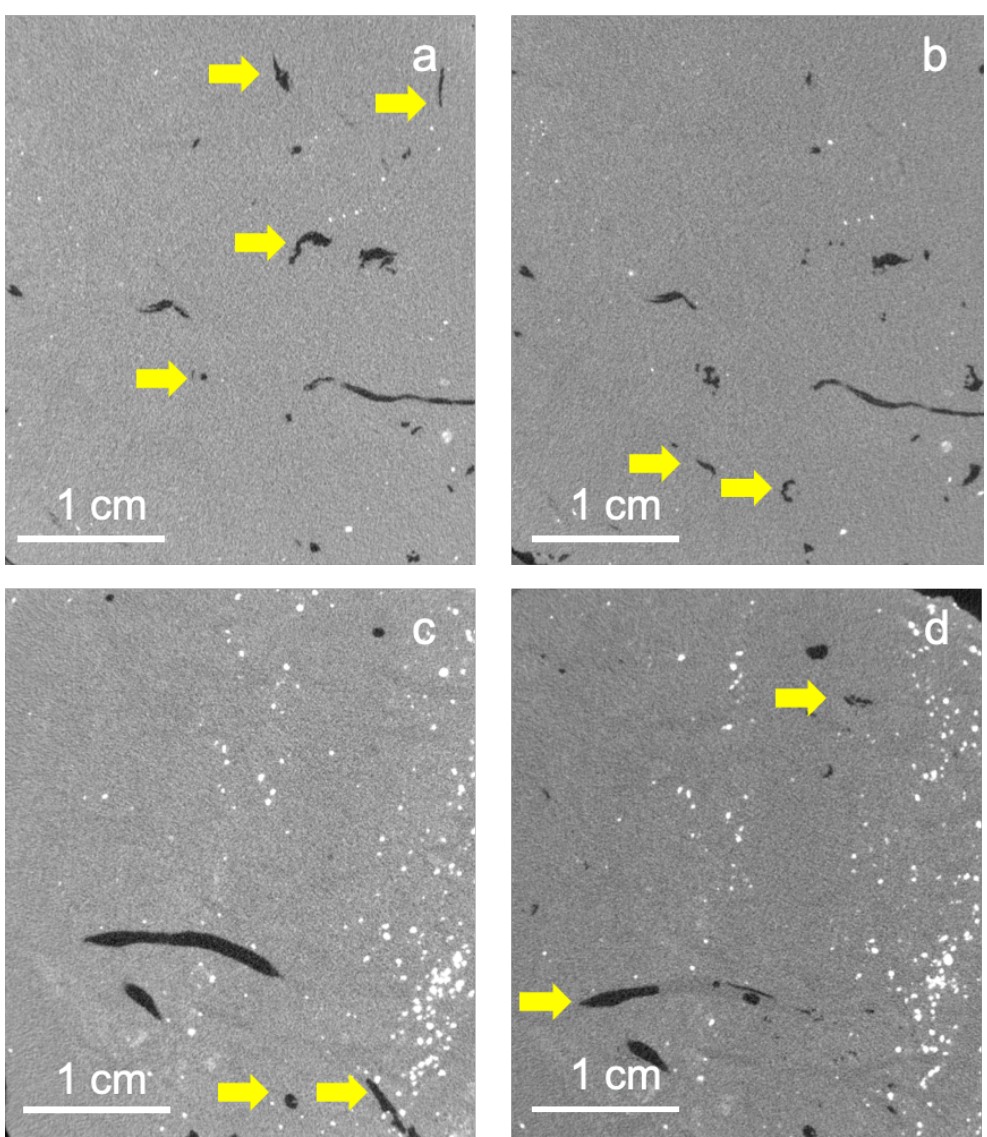

**Figure 2.** Pairs of 2D μ-CT images of samples A (**a** and **b**) and B (**c** and **d**) extracted from the 3D X-ray μ-CT images in grey levels showing air filled pores (black), the solid organic material with water filling the associated pores (dark grey) and particles of iron oxy-hydroxides. The horizontal yellow arrows on the images before freezing (**a** and **c**) show pores that were not present after freezing (**b** and **d**) and those on the images after freezing (**b** and **d**) correspond to pores that were not present before freezing or highly different (**a** and **c**).


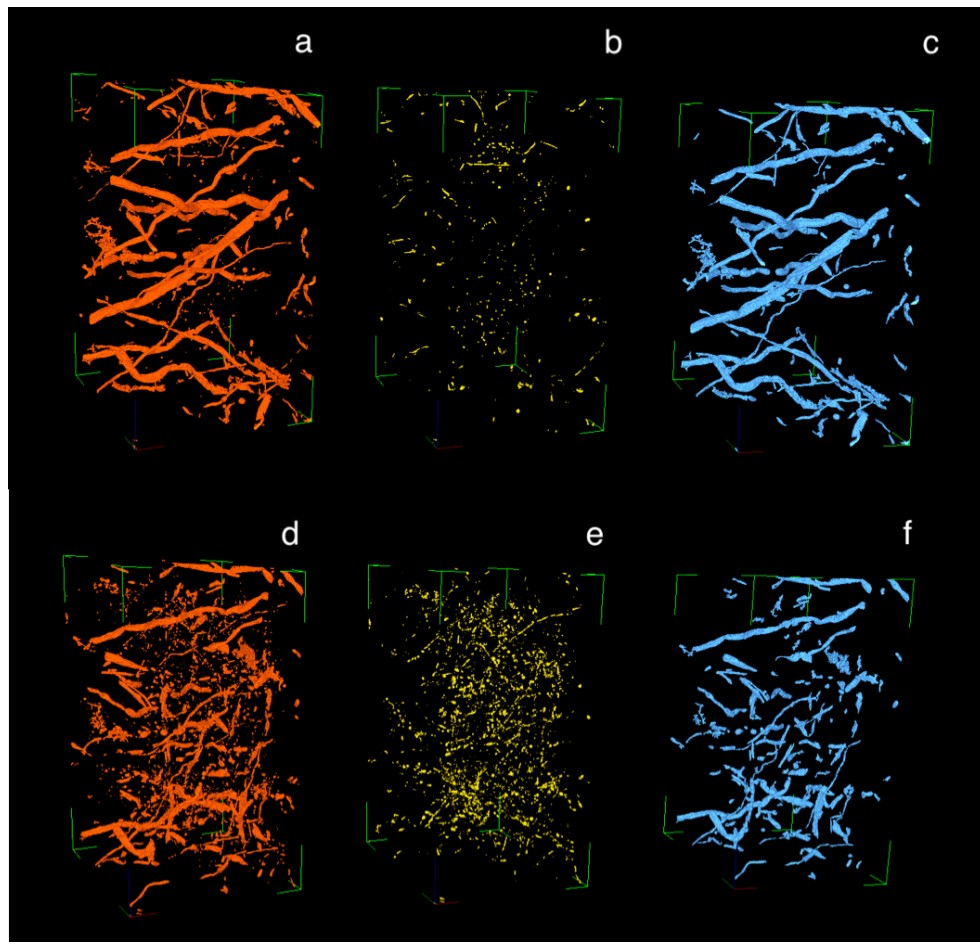


**Figure 3.** 3D X-ray μ-CT binary images 430×600×800 voxels in volume (~2.6×3.6×4.8 cm$^3$) of sample A showing the whole pores detected before (**a**) and after (**d**) freezing, the pores smaller than 500 voxels in volume before (**b**) and after (**e**) freezing, and the pores larger than 500 voxels in volume before (**c**) and after (**f**) freezing.





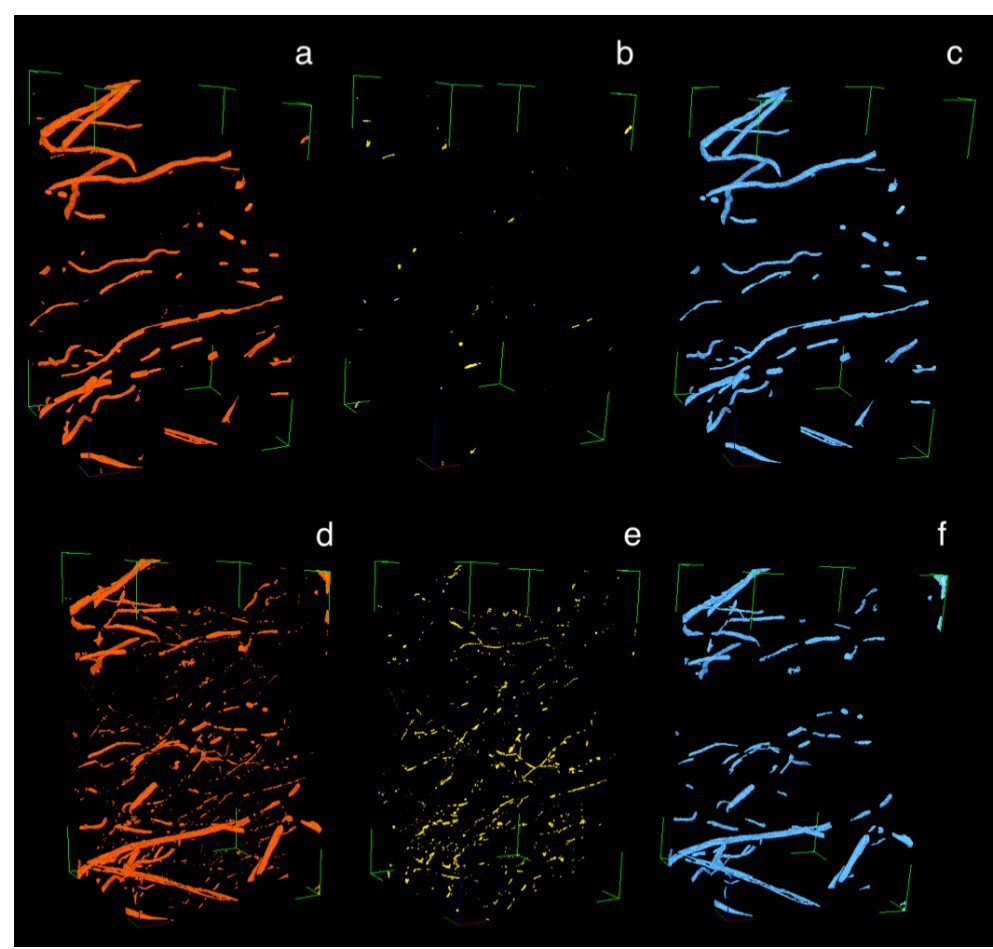

**Figure 4.** 3D X-ray μ-CT binary images 430×530×850 voxels (~2.6×3.2×5.1 cm$^3$) of sample B showing the whole pores detected before (**a**) and after (**d**) freezing, the pores smaller than 500 voxels in volume before (**b**) and after (**e**) freezing, and the pores larger than 500 voxels in volume before (**c**) and after (**f**) freezing.




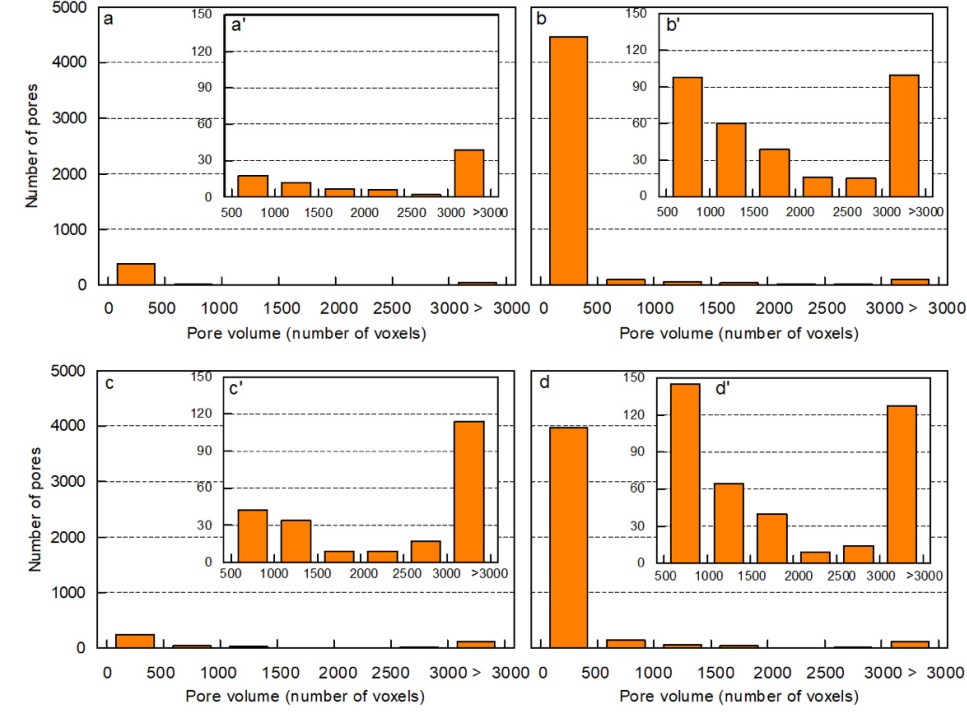


**Figure 5.** Pore volume distribution according to the number of voxels $60{\times}60{\times}60$ μm$^3$ in volume in the 3D X-ray μ-CT images of sample A before freezing (**a**, **a'**), and after freezing (**b**, **b'**), and of sample B before freezing (**c**, **c'**), and after freezing (**d**, **d'**).




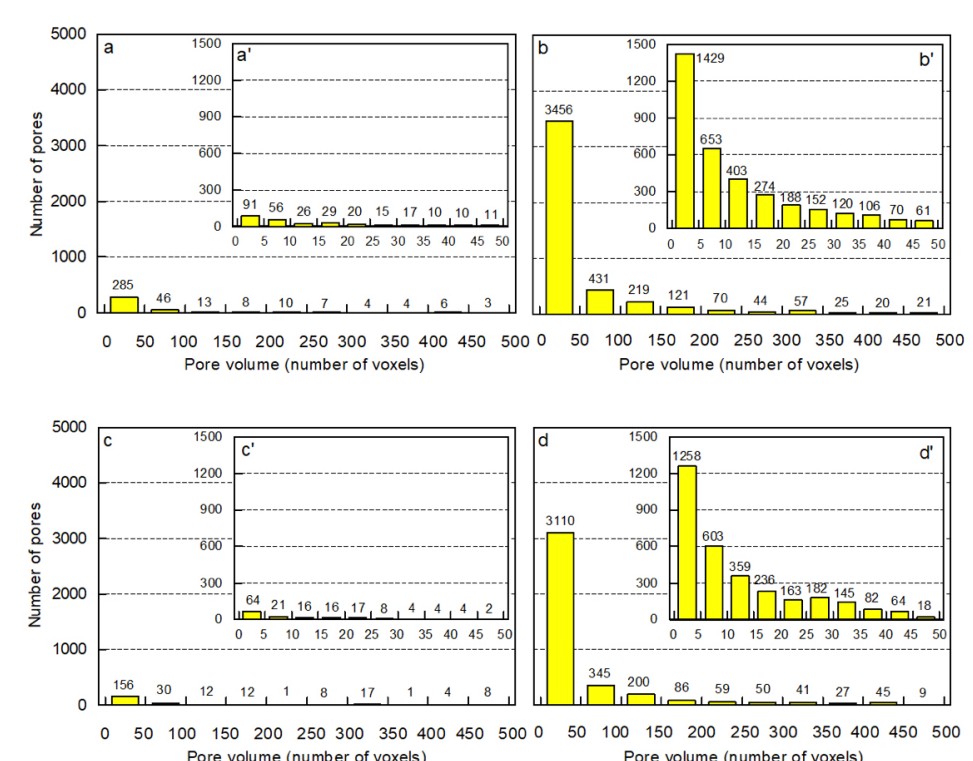


**Figure 6.** Pore volume distribution according to the number of voxels ($\leq$ 500) and ($\leq$ 50), 60×60×60 $\mu m^3$ in volume in the 3D X-ray $\mu$-CT images of sample A before freezing (**a**, **a'**), and after freezing (**b**, **b'**), and of sample B before freezing (**c**, **c'**), and after freezing (**d**, **d'**).





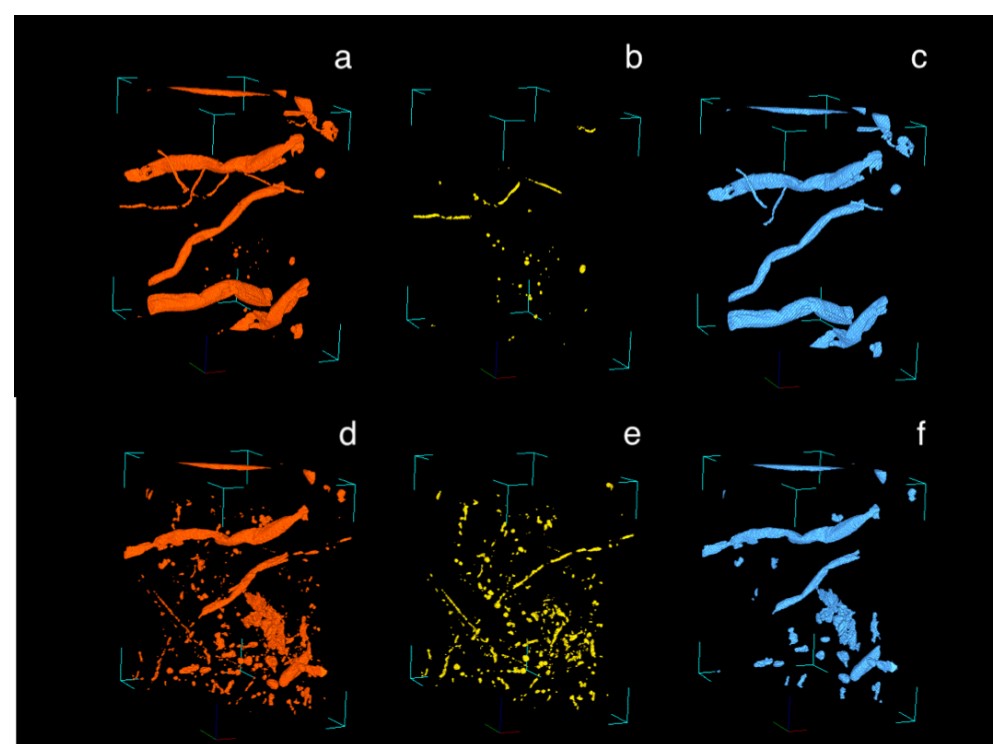


**Figure 7.** 3D X-ray μ-CT binary sub-images 200×350×350 voxels in volume (~1.2×2.1×2.1 cm$^3$) of sample A showing the whole pores detected before (**a**) and after (**d**) freezing, the pores smaller than 500 voxels in volume before (**b**) and after (**e**) freezing, and the pores larger than 500 voxels in volume before (**c**) and after (**f**) freezing.






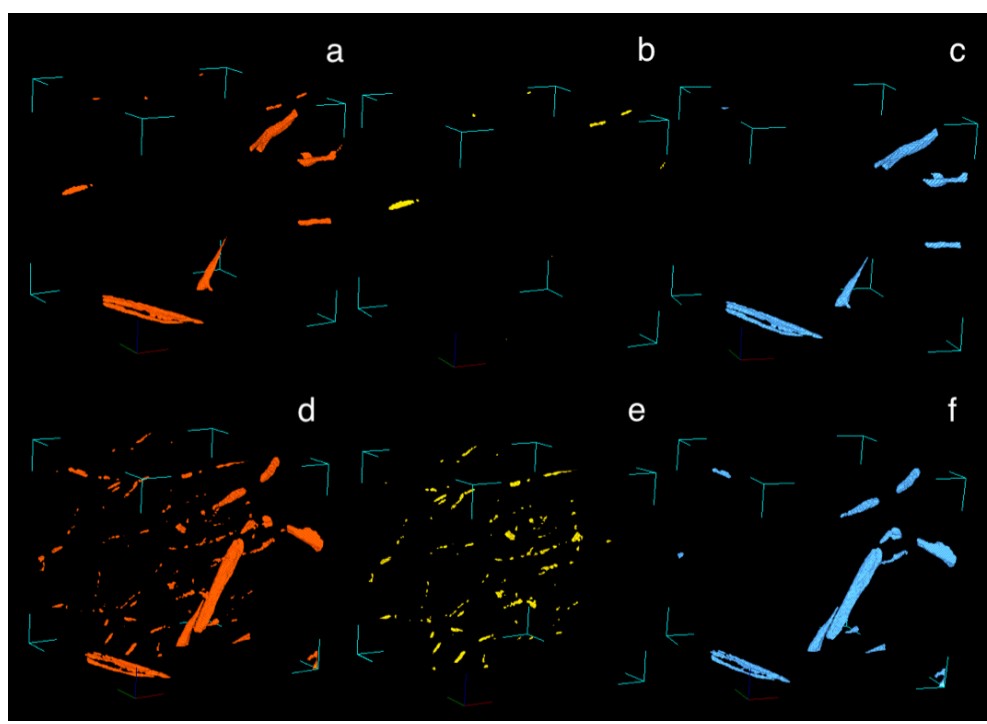

**Figure 8.** 3D X-ray μ-CT binary sub-images 300×300×300 voxels in volume (~1.8×1.8×1.8 cm$^3$) of sample B showing the whole pores detected before (**a**) and after (**d**) freezing, the pores smaller than 500 voxels in volume before (**b**) and after (**e**) freezing, and the pores larger than 500 voxels in volume before (**c**) and after (**f**) freezing.






**Table 1** Main physical and chemical characteristics of samples A and B of the highly decomposed peat material
studied.

| Sample | Depth (cm) | $\phi_T$ | $D_p$ | $\theta$ (cm$^3$ cm$^{-3}$) | $D_b$ (g cm$^3$) | $PPI$ | $C{:}N$ |
|---|---|---|---|---|---|---|---|
| A | 25-40 | 0.918 | 1.692 | 0.893 | 0.135 | 96.1 | 12.1 |
| B | 25-40 | 0.904 | 1.779 | 0.883 | 0.178 | 78.9 | 16.6 |

$\phi_T$: total porosity, $D_p$: particle density, $\theta$: water content at sampling, $D_b$: bulk density and PPI: pyrophosphate
index.






**Table 2** Characteristics of the pores in the 3D X-ray CT images of samples A and B of the highly decomposed
peat material studied before and after freezing.

| Sample | $\phi$ | Number of pores | Average size of the pores (voxels) |
|---|---|---|---|
| A before freezing | 0.009 | 470 | 3952 |
| A after freezing | 0.017 | 4792 | 732 |
| B before freezing | 0.005 | 474 | 2043 |
| B after freezing | 0.011 | 4371 | 488 |

*$\phi$: porosity measured in 3D X-ray CT*