# Peer review of "Effect of freezing on the microstructure of a highly decomposed peat material close to water saturation when used prior to X-ray micro computed tomography"

_SOIL, 2021_

## Referee Comment (RC1)

This study analyzed the structure of a peat material before and after freezing by investigating looking at the air-filled pore using X-ray μ-CT. Many scholars have studied the effects of freeze-thaw cycles on soil structure, and this study only focused on before and after freezing. Although the thesis of the study can attract our insights to read the paper, lots of questions still have not resolved clearly. Some suggestions should be useful to improve this manuscript.

1. Author seems to use only two samples for the experiment. Is the sample number enough to support the conclusion of the study?

2. Second, some methodological details are missing. Essential information about methods and materials is crucial for readers to evaluate the results. The description of the experimental process is not clear enough. What is the basis of freezing and thawing temperature selection? What container is the sample placed in when scanning?(line 121: a transparent plastic tube; line 137: were mounted and waxed on a glass rod)

3. Another major issue, the experimental procedure was not described in detail. This study only paid attention to air-filled pores, how to determine that the changes in soil structure before and after freezing are not the result of water change or migration.

4. What is the purpose of sub-images of the 3D images. This information is present in Figs. 3 and 4, which only slightly enlarged in the sub-images.

5. As for the results, we found that the upper and lower characteristic pores of the sample were relatively corresponding before and after freezing. Why did the middle part change so much.

6. The manuscript needs a major editorial revision to improve the writing quality. I can see some grammar mistakes, improper words, though I'm not a native speaker.

For example lines 30-34 "Theoretical calculation of the consequences of the increase in the specific volume of water by 8.7 % when it turns from liquid to solid because of freezing led to the creation of a pore volume in the organic matrix which remains saturated by water when returning to room temperature and consequently to the desaturation of the largest pores of the organic matrix as well as the finest tubular pores which were water-filled before freezing."

Lines 129-130 "X-ray u-CT was performed for the sub-samples 4x4x7 cm in volume cut between 30 and 37 cm depth using a micro X-ray u-CT device Nanotom 180NF (GE Phoenixx-ray, Wunstorf, Germany)." Suggest reorganizing the language.

---

## Author Comment (AC8)

**Effect of freezing prior to X-ray micro computed tomography on the microstructure of a highly decomposed peat material close to water saturation**

Hassan Al Majou[1,2], Ary Bruand[1], Olivier Rozenbaum[1,3], Emmanuel Le Trong[1]

[1] Université d'Orléans, CNRS, BRGM, Institut des Sciences de la Terre d'Orléans (ISTO), 1A rue de la Férollerie, 45071 Orléans Cedex 2 (France)

[2] University of Damas, Department of Soil Science, Faculty of Agronomy, PO Box 30621, Damas (Syria)

[3] CNRS, Conditions Extrêmes et Matériaux : Haute Température et Irradiation (CEMHTI), UPR 3079, 1 Avenue de la Recherche Scientifique, 45071 Orléans, Cedex 2 (France)

*Correspondence to:* Ary Bruand (Ary.Bruand@univ-orleans.fr)

**Abstract.** The modelling of peatland functioning, in particular the impact of anthropogenic warming and direct human disturbance on $CO_2$, $CH_4$ and $N_2O$, requires detailed knowledge of the peat structure and of both water and gas flow with respect to the groundwater table level. To this end, freezing is nowadays increasingly used to obtain small size peat samples for X-ray micro computed tomography (X-ray μ-CT) as required by the need to increase the resolution of the 3D X-ray CT images of the peat structure recorded. The aim of this study was to analyze the structure of a peat material before freezing and post-defreezing using X-ray μ-CT and to look for possible alterations in the structure by analyzing the air-filled porosity. A highly decomposed peat material close to water saturation was selected for study and collected between 25 and 40 cm depth. Two samples $4\times4\times7$ cm$^3$ in volume were analyzed before freezing and post-defreezing using an X-ray μ-CT Nanotom 180NF. Results showed that the continuity and cross section of the air-filled tubular pores several hundreds to about one thousand micrometers in diameter were altered post-defreezing. Many much smaller air-filled pores not detected before freezing were also recorded post-defreezing with 470 and 474 pores higher than one voxel in volume ($60\times60\times60$ μm$^3$ in volume each) before freezing, and 4792 and 4371 air-filled pores higher than one voxel in volume post-defreezing for the two samples studied. Detailed analysis showed that this increase resulted from a difference in the whole range of pore size studied and particularly from a dramatic increase in the number of air-filled pores ranging between 1 voxel ($216 \ 10^3$ μm$^3$) and 50 voxels ($10.8 \ 10^6$ μm$^3$) in volume. The increase in the specific volume of water by 8.7% when it turns from liquid to solid because of freezing led to the creation of a pore volume in the organic matrix which remained saturated by water when returning to room temperature. This induced the desaturation of some of the finest tubular pores as well as some the largest pores of the porous organic matrix which were both water-filled before freezing. The volume of these pores newly occupied by air using X-ray μ-CT and their total volume was found to be consistent with the one calculated as resulting from the increase in the specific volume of water when it turns into ice. They correspond to small air-filled pores several voxels in volume to several dozen voxels in volume and to discontinuous air-filled fine tubular pores which were both detected only post-defreezing. Finally, the increase in the specific volume of water because of freezing and related increase in the porosity of the water-
saturated porous matrix also appear also to be responsible for the alteration of the air-filled tubular pores detected

[revised manuscript text omitted]
). A highly decomposed peat was selected for study because it is potentially highly sensitive to the creation of structure artifacts during freezing due to its high water content and low fiber content. Large undisturbed samples (15×15×25 cm$^3$) were collected between 25 and 40 cm depth to avoid the heterogeneity of the top 20 cm due to *Molinia* roots. They were collected when the groundwater table level was close to the soil surface. The samples were stored at 3–4 °C to limit biological activity and in sealed plastic bags to avoid water loss.

**2.2 Physico-chemical analysis**

Bulk density and particle density were determined by using undisturbed peat samples a dozen cubic centimeters in volume and the kerozene method developed by Monnier et al. (1973). The total porosity was obtained by dividing the volume of water contained in a saturated sample by the known volume of the sample as described by Boelter (1976) and Nimmo (2013). The water content of the collected samples was determined after oven drying at 105°C for 24h. The degree of peat decomposition was characterized with the pyrophosphate index (Kaila 1956) which was determined following Gobat et al. (1986). The C and N contents were determined by combustion of dried and crushed samples at 1100°C, using a CNS-2000 LECO apparatus.

**2.3 Sub-sampling in the laboratory**

In order to have samples of the appropriate size for X-ray µ-CT, sub-samples of peat materials $4 \times 4 \times 7$ cm³ in volume corresponding to the depth of 30–37 cm were prepared by cutting with a scalpel blade to limit disturbance of the peat structure as far as possible. Then, each sample was placed in a transparent PVC tube 5 cm in diameter which was then hermetically sealed with a screw cap to avoid water loss. They were first imaged by X-ray µ-CT and then, on the basis of the methodology developed and used by Rezanezhad et al. (2010), Ramirez et al. (2016) and Moore et al. (2017), they were frozen at -10°C for 48 h, defrosted for 48 h at 20°C and imaged again by X-ray µ-CT. Each sealed PVC tube with its peat material was weighed at the different steps of the process, i.e. prior and after each X-Ray µ-CT imaging, to check the absence of water loss during the acquisition of the projected 2D images. Measurements showed that the weight variation between two successive steps and between the first and last step was <0.03 g for the two samples studied. This amount loss was considered negligible.

**2.4 X-ray Computed Tomography imaging (2D and 3D images)**

X-ray µ-CT was performed for the sub-samples $4 \times 4 \times 7$ cm³ in volume using a micro X-ray µ-CT device Nanotom 180NF (GE Phoenix|x-ray, Wunstorf, Germany). This equipment has a 180-kV nanofocus X-ray tube and a digital detector array ($2304 \times 1152$ pixels, Hamamatsu detector). Samples were placed in the chamber and rotated by 360 degrees during acquisition. The samples were centered and waxed on a sample holder (circular plate) whose axis of rotation was collinear to that of the tomograph chuck. An operating voltage of 120 kV and a filament current of 100 µA were applied. The distance between the X-ray source and the sample and between the X-ray source and the detector was 300 and 500 mm, respectively, giving a voxel size of 60 µm. The tomograph detector recorded 2D projections in 16 bit, i.e. divided into 65536 grey levels. The resulting projections were converted into a 3D image stack using a microcluster of four personal computers (PCs) with the Phoenix 3D reconstruction software. A filtered backprojection algorithm was used according to Feldkamp et al. (1984). The reconstruction software contained several different modules for artifact reduction (beam hardening, ring artifacts) to optimize the results. After reconstruction, the images were recorded in 8 bit (256 levels of grey) by always checking that the histograms of the 16-bit and 8-bit images were similar and that the images visually indistinguishable. During this process, we increased the dynamic range of the image by spreading the histogram over the entire range between 0 and 255 levels of grey. This facilitated the subsequent segmentation step. The 2000 projection images (angular increment of 0.18°) were acquired during sample rotation (with an acquisition time of 4 hours) for every sample before freezing and post-defreezing. As the cone beam geometry created artifacts, the first and the last 76 cross-sectional images were removed (Le Trong et al., 2008; Rozenbaum and Rolland du Roscoat, 2014).

The resulting 3D images were cropped for sample A to a size of 430×600×800 voxels corresponding to

2.6×3.6×4.8 cm$^3$ before freezing and post-defreezing, and for sample B to a size of 430×530×850 voxels before freezing and post-defreezing corresponding to 2.6×3.2×5.1 cm$^3$, each image in a local 3D coordinate system with a voxel size of 60×60×60 μm$^3$ for samples A and B before freezing and post-defreezing.

**2.5 X-ray image analysis (segmentation and attenuation)**

A region of interest that excluded the irregular sample boundaries and outside region was defined for every sample by identifying the largest rectangular parallelepiped image in the cylindrical sample studied. Smoothing the 3D

images with a moving average filter over a window of 5×5×5 voxels increased their signal-to-noise ratios from the range [7.6-9.2] to [12.5-15.0]. The moving average filter replaced each voxel of the original image by the average grey value of its neighbors over a window centered on it as this is considered the optimal method to remove random noise on grey-level images (Smith, 1997). The signal-to-noise ratio was computed as being the ratio of the average of the grey levels of the image to their standard deviation according to Avcibas et al. (2002). They were then segmented by thresholding. The threshold value used was the absolute minimum between the two peaks of the bimodal distribution of the grey levels of the voxels of each image (Fig. 1) (Rozenbaum et al., 2012). The grey level corresponding to the threshold value for sample A before freezing and post-defreezing was 93 and 78

(Fig. 1a), respectively, and for sample B before freezing and post-defreezing, it was 80 and 68, respectively (Fig.

1b). The voxels with a grey level smaller than the threshold value were considered as being pore voxels while those with a grey level higher than or equal to the threshold value were considered as matrix voxels. This simple procedure has no adjustable parameter and therefore introduces no bias when comparing the images. In each binary image, each pore (i.e. group of contiguous foreground voxels surrounded by matrix voxels) was identified by a voxel-by-voxel scanning of the image, and its volume (in terms of number of voxels) recorded by the following algorithmic procedure:

- Consider each voxel during a raster scan of the image. Let *v* be the current voxel;

- If *v* is a matrix voxel, or has been marked as belonging to an already identified pore, proceed to the next voxel;

- If *v* is a pore voxel belonging to a yet unidentified pore, starting from *v*, perform a geodesic reconstruction of the pore (Lantuéjoul and Beucher, 1981). During the reconstruction of the pore, mark all its voxels as belonging to an identified pore, and keep count of their number. Once the reconstruction is complete, the number of voxels yields the volume of the pore. Proceed to the next voxel in the raster scan.

The outcome of this procedure is list of pores, of their volume and their position in the 3D image. The total porosity is then the ratio of the total pore volume divided by the total volume of the rectangular parallelepiped selected.

**3 Results and discussion**

**3.1 Characteristics of the peat samples studied**

The measured physical characteristics of the peat samples studied are given in Table 1. The volumetric water contents at sampling were similar for the two samples A and B (0.893 and 0.883 $cm^3$ $cm^{-3}$, respectively). These values are much higher than those recorded by Rezanezhad et al. (2010) for sphagnum peat materials (between

0.38 and 0.43 $cm^3$ $cm^{-3}$) collected between the surface and 67 cm depth. Their peat materials were collected far from water saturation because the groundwater table level was far from the surface, whereas our samples were collected with a groundwater table level close to the soil surface. The porosity values of samples A and B (0.918

and 0.904, respectively) are close to their water content, thus indicating that they are close to water saturation. The measured bulk densities recorded for the two samples A and B (0.135 to 0.178 g $cm^{-3}$, respectively) are consistent with those of highly decomposed peat materials (Benscoter et al., 2011; Kurnain and Hayati, 2016). The measured pyrophosphate indices recorded for samples A and B (96.1 and 78.9, respectively) are also consistent with highly decomposed peat materials which can be classified as asapric peat (pyrophosphate index >30) according to

Levesque et al. (1980). The C/N ratio recorded for samples A and B (12.1 and 16.6, respectively) confirms that the two peat samples present a high degree of decomposition (Comont et al. 2006). Finally, the dry bulk density values recorded for the two samples A and B (0.135 and 0.178 g $cm^{-3}$, respectively) are much closer to the values recorded for a well-decomposed peat material resulting from *Sphagnum* moss with a fiber content of only 15%

(0.250 g $cm^{-3}$) than to the values recorded for undecomposed peat materials with a fiber content of 98% (0.009 g

$cm^{-3}$) (Boelter, 1968). Overall, the peat material selected for this study is much more decomposed than the peat materials studied by Quinton et al. (2009) and Rezanezhad et al. (2010).

**3.2 Comparison of the 2D and 3D X-ray μ-CT images in grey levels before freezing and post-defreezing**

The same heights were chosen for the 3D X-ray μ-CT images in grey levels before freezing and post-defreezing for samples A (800 voxels) and B (850 voxels). The final image sizes chosen were then 430×600×800 voxels (~2.6×3.6×4.8 $cm^3$) before freezing and post-defreezing for sample A and 430×530×850 voxels (~2.6×3.2×5.1 $cm^3$) before freezing and post-defreezing for sample B.

[revised manuscript text omitted]

Sub-images of the 3D X-ray μ-CT images recorded were selected to compare the difference in pore morphology qualitatively before freezing and post-defreezing more easily than with the whole images in which the high number of pores limited the morphological analysis (Figs. 3 and 4), particularly for the pores <500 voxels (Figs. 3b and e,

4b and e). Thus, one 3D X-ray μ-CT sub-image 200×350×350 voxels in volume (~1.2×2.1×2.1 cm$^3$) and another one 300×300×300 voxels in volume (~1.8×1.8×1.8 cm$^3$) were selected for samples A and B, respectively (Figs. 7

and 8). The selected sub-images showed that the pores <500 voxels corresponded to air-filled pores of several voxels to several dozen voxels, and to discontinuous air-filled fine tubular pores (Figs. 7b and e, 8b and e). The comparison showed that freezing led to a dramatic increase in the number of air-filled pores (Figs. 7 and 8) and to the appearance or disappearance of discontinuous air-filled fine tubular pores <500 voxels (Figs. 7b and e, 8b and e).

**3.4 Origin of the difference recorded before freezing and post-defreezing**

As freezing leads to an 8.7% increase in the specific volume of the water, the possible consequences of this increase on the changes recorded for the peat material studied were analyzed. The total porosity before freezing ($\phi_{T, BF}$) can be written as follows:

$\phi_{T,BF} = V_{V, BF} / (V_S + V_{V,BF})$ (2)

where $V_{V, BF}$ is the total specific volume of pores of the peat material before freezing in cm$^3$ g$^{-1}$, $V_S$ is the specific volume of the organic solid phase dried at 105°C in cm$^3$ g$^{-1}$ and equal to 0.591 cm$^3$ g$^{-1}$ and 0.562 cm$^3$ g$^{-1}$ for samples A and B, respectively (reciprocal of the particle density measured for peat materials A and B) (Table 1).

Thus, using equation (2):

$V_{V, BF} = \phi_{T, BF} \times V_S / (1 - \phi_{T, BF})$ (3)

which gives $V_{V, BF}$ = 6.616 cm$^3$ g$^{-1}$ and 5.292 cm$^3$ g$^{-1}$ for samples A and B, respectively. The specific volume of pores before freezing, $V_{V, BF}$, can be decomposed as follows:

$V_{V, BF} = V_{V, Mwf, BF} + V_{V, TPwf, BF} + V_{V, TPaf, BF}$ (4)

where $V_{V, Mwf, BF}$ is the specific volume of pores of the organic matrix saturated with water before freezing in cm$^3$ g$^{-1}$

$V_{V, TPwf, BF}$ is the specific volume of tubular pores occupied with water before freezing in cm$^3$ g$^{-1}$ and $V_{V, TPaf, BF}$ is the specific volume of tubular air-filled pores before freezing in cm$^3$ g$^{-1}$. The porosity related to the tubular air- filled pores before freezing, $\phi_{TPaf, BF}$, in the whole peat material is:

$\phi_{TPaf, BF} = V_{V, TPaf, BF} / (V_S + V_{V, BF})$ (5)

Thus:

$V_{V, TPaf, BF} = \phi_{TPaf, BF} \times (V_S + V_{V, BF})$ (6)

which gives $V_{V, TPaf, BF}$ = 0.065 cm$^3$ g$^{-1}$ and 0.029 cm$^3$ g$^{-1}$ for samples A and B, respectively, with the values of

$\phi_{TPaf, BF}$ corresponding to the value of $\phi$ measured before freezing using X-ray μ-CT (Table 2).

The increase in the specific volume of liquid water by 8.7% when it turns to solid from 20°C to –10°C (Harvey,

2017) increases the porosity of both the pores in the organic matrix and the tubular pores fillled with water before freezing to a porosity post-defreezing which can be calculated as follows:

$\phi_{Mwf, AF} + \phi_{TPwf, AF} = [(V_{V, Mwf, BF} + V_{V, TPwf, BF}) \times 1.087] / [V_S + ((V_{V, Mwf, BF} + V_{V, TPwf, BF}) \times 1.087) + V_{V, TPaf, BF}]$ (7)

where $\phi_{Mwf, AF}$ is the porosity related to the water-filled of the organic matrix post-defreezing, $\phi_{TPwf, AF}$ is the porosity related to the water-filled tubular pores post-defreezing, and 1.087 the coefficient by which the volume of water is increased when it turns from liquid (20°C) to solid (–10°C). Using equation (4):

$\phi_{Mwf, AF} + \phi_{TPwf, AF} = [(V_{V, BF} - V_{V, TPaf, BF}) \times 1.087] / [V_S + ((V_{V, BF} - V_{V, TPaf, BF}) \times 1.087) + V_{V, TPaf, BF}]$        (8)

which gives $\phi_{Mwf, AF} + \phi_{TPwf, AF}$ = 0.916 and 0.906 for peat materials A and B, respectively by using values of $V_{V, BF}$

and $V_{V, TPaf, BF}$ given by equations (3) and (6), respectively. These values can be compared to those of $\phi_{Mwf, BF}$ +

$\phi_{TPwf, BF}$ before freezing that were calculated as follows:

$\phi_{Mwf, BF} + \phi_{TPwf, BF} = \phi_T - \phi_{TPaf, BF}$                                                     (9)

with $\phi_{TPaf, BF}$ equal to $\phi$ before freezing (Table 2), which gives $\phi_{Mwf, BF} + \phi_{TPwf, BF}$ = 0.909 and 0.899 for samples A

and B, respectively. Thus, according to these results and the related assumptions, the increase in the water-filled porosity of samples A and B post-defreezing was 0.007. These values can be compared with the increase in the air-filled porosity and measured with the 3D X-ray μ-CT images recorded in this study. This increase post- defreezing was 0.008 and 0.006 for samples A and B, respectively and thus similar to the calculated values.

Based on these different results, two scenarios can be proposed to explain what happened during the freeze-thaw process. According to the first scenario, during freezing, the water-filled pore volume corresponding to the sum of the pores of the organic matrix and of fine tubular pores increases by 8.7% because of the increase in the specific volume of water when it turns from liquid to solid. After thawing to room temperature, the peat material keeps the memory of this evolution during the freezing phase. As the specific pore volume of the highly decomposed organic matrix increases in volume following the formation of ice, it does not then decrease after thawing, with the result that the water, which is located preferentially in the smallest pores, small tubular pores and largest pores of the organic matrix, all saturated with water before freezing, is no longer located in these pores when the water turns from solid to liquid after thawing. The porosity newly occupied by air was measured by using 3D X-ray μ-CT

(Table 2) and corresponds to the increase in porosity calculated following the transformation of liquid water into ice. The second scenario assumes that most of the small elongated pores would be already partially air-filled but their volumes are too small to be detected with the 3D X-ray μ-CT used in this study. After the freeze-thaw process for the same reasons as for the first scenario concerning the evolution of the porous organic matrix, the proportion and size of small air-filled pores increases, making them detectable by 3D X-ray μ-CT.

Finally, the increase in the specific volume of water because of freezing may also be responsible for the alteration
of the already air-filled tubular pores >500 voxels before freezing as shown by the 3D binary images (Figs. 3 and
4) and the pore volume distribution (Figs. 5 and 6) because of deformations in the structure of the surrounding
porous organic matrix during freezing.

**4. Conclusions**

Our results show that the freezing technique that can be used prior to peat material sub-sampling as required by
3D X-ray μ-CT altered the structure of the highly decomposed and close to water saturation peat material studied.
Both the continuity and cross section of the tubular pores measuring from several hundreds to about one thousand
micrometers in diameter differed before freezing and post-defreezing. These pores were several hundreds to
several thousand voxels in volume in the 40 cm$^3$ highly decomposed peat material studied, one voxel
corresponding to 216 μm$^3$ in volume. Results also show that very small air-filled pores several voxels to several
dozen voxels in volume and discontinuous air-filled fine tubular pores within the peat material studied were only
detected post-defreezing in the samples. The increase in the specific volume of water by 8.7% when it turns from
liquid to solid because of freezing led to the creation of a pore volume in the organic matrix which remained
saturated by water when returning to room temperature. This induced the desaturation of some of the finest tubular
pores as well as some the largest pores of the porous organic matrix which were both water-filled before freezing.
The volume of these pores newly occupied by air post-defreezing was measured using X-ray μ-CT and their
cumulated volume was found to be consistent with the one calculated by taking into account the thermal expansion
of water from 20°C (liquid) to –10°C (ice). We conclude that the increase in the specific volume of water because
of freezing is also responsible for the alteration of the already air-filled tubular pores >500 voxels before freezing,
as shown by the 3D binary images and the pore volume distribution, and that this alteration is a consequence of
the deformation of the organic matrix due to the increase in the specific volume of water when it turns from liquid
to solid because of freezing. Finally, our results show clearly that both the pore morphology and pore size
distribution, and more globally the structure of the highly decomposed peat material studied, were altered by
freezing. Thus, the possible consequences of freezing prior to any study of the structure of peat materials should
be investigated, particularly for highly decomposed peat materials. Future work will focus on the possible presence
of pore geometry artifacts similar to those recorded in our study in less decomposed peat materials, which may be
less sensitive to the occurrence of artefacts post-defreezing.

[revised manuscript text omitted]

Figure 2. Pairs of 2D µ-CT images of samples A (**a** and **b**) and B (**c** and **d**) extracted from the 3D X-ray µ-CT images in grey levels showing air filled pores (very dark grey), the solid organic material with water filling the associated pores (dark grey) and particles of iron oxy-hydroxides (very light grey). The numbers identify pores which were present before freezing (**a** and **c**) and still present post-defreezing (**b** and **d**) but with a different shape or size (blue), present before freezing and not post-defreezing (red) and not present before freezing but present post-defreezing (yellow).

[Figure]

**Figure 3.** 3D X-ray μ-CT binary images 430×600×800 voxels in volume (~2.6×3.6×4.8 cm$^3$) of sample A showing
the whole pores detected before freezing (**a**) and post-defreezing (**d**), the pores smaller than 500 voxels in volume
before freezing (**b**) and post-defreezing (**e**), and the pores larger than 500 voxels in volume before freezing (**c**) and
post-defreezing (**f**).

[Figure]

**Figure 4.** 3D X-ray μ-CT binary images 430×530×850 voxels (~2.6×3.2×5.1 cm$^3$) of sample B showing the whole pores detected before freezing (**a**) and post-defreezing (**d**), the pores smaller than 500 voxels in volume before freezing (**b**) and post-defreezing (**e**), and the pores larger than 500 voxels in volume before freezing (**c**) and post-defreezing (**f**).

[Figure]

**Figure 5.** Pore volume distribution according to the number of voxels $60×60×60$ $\mu m^3$ in volume in the 3D X-ray

$\mu$-CT images of sample A before freezing (**a**, **a'**), and post-defreezing (**b**, **b'**), and of sample B before freezing (**c**,

**c'**), and post-defreezing (**d**, **d'**).

[Figure]

**Figure 6.** Pore volume distribution according to the number of voxels ($\leq 500$) and ($\leq 50$), $60 \times 60 \times 60$ µm³ in volume in the 3D X-ray µ-CT images of sample A before freezing (**a**, **a'**), and post-defreezing (**b**, **b'**), and of sample B before freezing (**c**, **c'**), and post-defreezing (**d**, **d'**).

[Figure]

**Figure 7.** 3D X-ray μ-CT binary sub-images 200×350×350 voxels in volume (~1.2×2.1×2.1 cm³) of sample A
showing all the pores detected before freezing (**a**) and post-defreezing (**d**), the pores smaller than 500 voxels in
volume before freezing (**b**) and post-freezing (**e**), and the pores larger than 500 voxels in volume before freezing
(**c**) and post-freezing (**f**).

[Figure]

**Figure 8.** 3D X-ray μ-CT binary sub-images 300×300×300 voxels in volume (~1.8×1.8×1.8 cm³) of sample B
showing all the pores detected before freezing (**a**) and post-frteezing (**d**), the pores smaller than 500 voxels in
volume before freezing (**b**) and post-defreezing (**e**), and the pores larger than 500 voxels in volume before freezing
(**c**) and post-freezing (**f**).

**Table 1** Main physical and chemical characteristics of samples A and B of the highly decomposed peat material studied.

| Sample | Depth (cm) | $\phi_T$ | $D_p$ | $\theta$ (cm$^3$ cm$^{-3}$) | $D_b$ (g cm$^3$) | *PPI* | *C:N* |
|--------|------------|----------|-------|---------------------------|------------------|-------|-------|
| A | 25-40 | 0.918 | 1.692 | 0.893 | 0.135 | 96.1 | 12.1 |
| B | 25-40 | 0.904 | 1.779 | 0.883 | 0.178 | 78.9 | 16.6 |

*$\phi_T$: total porosity, $D_P$: particle density, $\theta$: water content at sampling, $D_b$: bulk density and PPI: pyrophosphate index.*

**Table 2** Characteristics of the pores in the 3D X-ray CT images of samples A and B of the highly decomposed peat material studied before freezing and post-defreezing.

| Sample | $\phi$ | Number of pores | Average size of the pores (voxels) |
|--------|--------|-----------------|-------------------------------------|
| A before freezing | 0.009 | 470 | 3952 |
| A post-defreezing | 0.017 | 4792 | 732 |
| B before freezing | 0.005 | 474 | 2043 |
| B post-defreezing | 0.011 | 4371 | 488 |

*$\phi$: porosity measured in 3D X-ray CT*